# Interactions between the Parasite *Philasterides dicentrarchi* and the Immune System of the Turbot *Scophthalmus maximus*. A Transcriptomic Analysis

**DOI:** 10.3390/biology9100337

**Published:** 2020-10-15

**Authors:** Alejandra Valle, José Manuel Leiro, Patricia Pereiro, Antonio Figueras, Beatriz Novoa, Ron P. H. Dirks, Jesús Lamas

**Affiliations:** 1Department of Fundamental Biology, Institute of Aquaculture, Campus Vida, University of Santiago de Compostela, 15782 Santiago de Compostela, Spain; alejandra.valle.cao@usc.es; 2Department of Microbiology and Parasitology, Laboratory of Parasitology, Institute of Research on Chemical and Biological Analysis, Campus Vida, University of Santiago de Compostela, 15782 Santiago de Compostela, Spain; josemanuel.leiro@usc.es; 3Institute of Marine Research, Consejo Superior de Investigaciones Científicas-CSIC, 36208 Vigo, Spain; patriciapereiro@iim.csic.es (P.P.); antoniofigueras@iim.csic.es (A.F.); beatriznovoa@iim.csic.es (B.N.); 4Future Genomics Technologies, Leiden BioScience Park, 2333 BE Leiden, The Netherlands; dirks@futuregenomics.tech

**Keywords:** *Philasterides dicentrarchi*, turbot, immune response, transcriptomics, infection

## Abstract

**Simple Summary:**

*Philasterides dicentrarchi* is a free-living ciliate that causes high mortality in marine cultured fish, particularly flatfish, and in fish kept in aquaria. At present, there is still no clear picture of what makes this ciliate a fish pathogen and what makes fish resistant to this ciliate. In the present study, we used transcriptomic techniques to evaluate the interactions between *P. dicentrarchi* and turbot leucocytes during the early stages of infection. The findings enabled us to identify some parasite genes/proteins that may be involved in virulence and host resistance, some of which may be good candidates for inclusion in fish vaccines. Infected fish responded to infection by generating a very potent inflammatory response, indicating that the fish use all of the protective mechanisms available to prevent entry of the parasite. The findings also provide some valuable insight into how the acute inflammatory response occurs in fish.

**Abstract:**

The present study analyses the interactions between *Philasterides dicentrarchi* (a ciliate parasite that causes high mortalities in cultured flatfish) and the peritoneal cells of the turbot *Scophthalmus maximus* during an experimental infection. The transcriptomic response was evaluated in the parasites and in the fish peritoneal cells, at 1, 2 and 4 h post-infection (hpi) in turbot injected intraperitoneally (ip) with 10^7^ ciliates and at 12 and 48 hpi in turbot injected ip with 10^5^ ciliates. Numerous genes were differentially expressed (DE) in *P. dicentrarchi*, relative to their expression in control ciliates (0 hpi): 407 (369 were up-regulated) at 1 hpi, 769 (415 were up-regulated) at 2 hpi and 507 (119 were up-regulated) at 4 hpi. Gene ontology (GO) analysis of the DE genes showed that the most representative categories of biological processes affected at 1, 2 and 4 hpi were biosynthetic processes, catabolic processes, biogenesis, proteolysis and transmembrane transport. Twelve genes of the ABC transporter family and eight genes of the leishmanolysin family were DE at 1, 2 and 4 hpi. Most of these genes were strongly up-regulated (UR), suggesting that they are involved in *P. dicentrarchi* infection. A third group of UR genes included several genes related to ribosome biogenesis, DNA transcription and RNA translation. However, expression of tubulins and tubulin associated proteins, such as kinesins or dyneins, which play key roles in ciliate division and movement, was down-regulated (DR). Similarly, genes that coded for lysosomal proteins or that participate in the cell cycle mitotic control, glycolysis, the Krebs cycle and/or in the electron transport chain were also DR. The transcriptomic analysis also revealed that in contrast to many parasites, which passively evade the host immune system, *P. dicentrarchi* strongly stimulated turbot peritoneal cells. Many genes related to inflammation were DE in peritoneal cells at 1, 2 and 4 hpi. However, the response was much lower at 12 hpi and almost disappeared completely at 48 hpi in fish that were able to kill *P. dicentrarchi* during the first few hpi. The genes that were DE at 1, 2 and 4 hpi were mainly related to the apoptotic process, the immune response, the Fc-epsilon receptor signalling pathway, the innate immune response, cell adhesion, cell surface receptors, the NF-kappaB signalling pathway and the MAPK cascade. Expression of toll-like receptors 2, 5 and 13 and of several components of NF-κB, MAPK and JAK/STAT signalling pathways was UR in the turbot peritoneal cells. Genes expressing chemokines and chemokine receptors, genes involved in prostaglandin and leukotriene synthesis, prostaglandins, leukotriene receptors, proinflammatory cytokines and genes involved in apoptosis were strongly UR during the first four hours of infection. However, expression of anti-inflammatory cytokines such as Il-10 and lipoxygenases with anti-inflammatory activity (i.e., *arachidonate 15-lipoxygenase*) were only UR at 12 and/or 48 hpi, indicating an anti-inflammatory state in these groups of fish. In conclusion, the present study shows the regulation of several genes in *P. dicentrarchi* during the early stages of infection, some of which probably play important roles in this process. The infection induced a potent acute inflammatory response, and many inflammatory genes were regulated in peritoneal cells, showing that the turbot uses all the protective mechanisms it has available to prevent the entry of the parasite.

## 1. Introduction

The subclass Scuticociliatia (Protozoa, Ciliophora) includes several families of free-living ciliates that are abundant in marine habitats. Some scuticociliate species, including *P. dicentrarchi*, have acquired the capacity to infect fish, causing high mortalities in some cultured fish species, including flatfish [1,2,3,4,5]. However, there is no clear picture of what makes this scuticociliate a fish pathogen. It is generally thought that *P. dicentrarchi* enters the fish through external lesions and then proliferates in the blood and in most internal organs where it causes important histopathological changes [1,3,6,7,8,9]. Although the mechanisms that *P. dicentrarchi* uses to invade fish tissues are not known, it has been suggested that during infection the ciliate can release proteases, which are considered virulence factors [10]. Ciliate proteases can destroy components of the turbot humoral immune response [11,12] and modify fish leukocyte functions [13,14], thus providing a mechanism for circumventing the fish immune system. In addition, *P. dicentrarchi* has developed other mechanisms that appear to be important in relation to parasitism. For example, it possesses mitochondria with two respiratory pathways: the cytochrome pathway, which exists in all aerobic organisms, and an alternative, cyanide-insensitive oxidase pathway, which enables the ciliate to survive and proliferate under normoxic and hypoxic conditions [15] and thus to become adapted to differences in oxygen levels in the host and seawater. In addition, *P. dicentrarchi* has potent antioxidant defence mechanisms that may be important during infection, including several superoxide dismutases that help the ciliate resist the reactive oxygen species released by the host [16]. Finally, it has been suggested that *P. dicentrarchi* can release its extrusome contents to create a protective barrier against soluble factors of the host immune system [17]. 

Several components of *P. dicentrarchi* stimulate fish leucocytes, thereby increasing respiratory burst, degranulation and the expression of pro-inflammatory cytokines. Despite the high level of stimulation, the toxic substances produced by the fish leucocytes do not seem to be sufficient to kill the parasite [11,18]. However, a turbot NK-lysin (an effector molecule of cytotoxic lymphocytes) has been reported to have very high antiparasitic activity, thus directly affecting the viability of *P. dicentrarchi* [19]. The innate humoral and adaptive immune responses appear to be crucial in defending the fish against this parasite. In this respect, the complement system plays a key role in the fish defence against this pathogen, as it displays potent antiparasitic activity, especially when activated by the classical pathway [11,20]. In addition to complement, the fish coagulation system also plays an important role in immobilizing and killing the parasite [21].

Little is known about how the fish immune system provides protection against parasites or even how it recognises the parasites (particularly ciliates) or which immune signalling pathways are stimulated. Pardo et al. [22,23] investigated how *P. dicentrarchi* stimulates the immune system of fish, by analysing the changes in gene expression in organs of the turbot immune system (spleen, liver and kidney) during a *P. dicentrarchi* infection. These researchers found that many genes regulating substances involved in the immune response, including chemokines, chemotaxins, complement, immunoglobulins, major histocompatibility complex, interferon, lectins, cytochrome P450 and lysozyme, were differentially expressed and that the response was strongest in the spleen at 1- and 3- days post-infection. Most transcriptomic studies of the immune response in fish infected with ciliates have been carried out with ectoparasites such as *Cryptocaryon irritans* and *Ichthyophthirius multifiliis* [24,25,26,27], usually with an infection time longer than 12 h. In addition, previous transcriptomic studies have analysed the response in either the parasite or in the cells of the fish immune system, but not in both. The scuticociliate *P. dicentrarchi* can overcome the external barriers of fish and proliferate in the blood and in internal organs, causing a systemic infection. The initial interaction between the ciliate and fish immune system may be crucial to the success of infection. In the present study, we used transcriptomic technologies to analyse the interactions between *P. dicentrarchi* and the turbot immune system during the very early stages of infection. Parasites were injected intraperitoneally and the transcriptomic response was evaluated in both fish cells and parasites found in the peritoneal cavity at several different times. This infection model enabled us to obtain sufficient numbers of host cells and parasites for the analysis.

## 2. Material and Methods

### 2.1. Fish and Ethical Statement

Healthy turbot, *Scophthalmus maximus* (L.), weighing about 50 g, were obtained from a local fish farm and maintained in 250 L tanks with recirculating and aerated seawater at 18 °C and fed daily with commercial pellets. All experimental protocols carried out in the present study followed the European legislation (Directive 2010/63/EU) and the Spanish legislative requirements related to the use of animals for experimentation (RD 53/2013) and were approved by the Institutional Animal Care and Use Committee of the University of Santiago de Compostela (Spain) (Ethic code: AGL2017-83577) on 01-02-2019. Before any experimental manipulation, fish were anaesthetized by immersion in a 100 mg/L solution of MS-222 (tricaine methane sulfonate; Sigma-Aldrich, Madrid, Spain) in seawater. At the end of the experiments, the fish were fully anaesthetized before being killed by pithing. 

### 2.2. Parasites

Specimens of *Philasterides dicentrarchi* (isolate I1), obtained from experimentally infected turbot, were cultured as previously indicated [11]. Briefly, parasites were cultured at 18 °C in flasks containing L-15 Leibovitz medium with 10% heat-inactivated foetal bovine serum, lipids (lecithin and Tween 80), nucleosides and glucose. To prepare the parasites, the ciliates were collected from the flasks, centrifuged at 700× *g* for 5 min, washed twice in phosphate-buffered saline (PBS) and resuspended in the same buffer.

### 2.3. Experimental Infection with P. dicentrarchi

Immediately prior to the present study, and using the same groups of fish, we found that with an inoculum dose of 10^7^ parasites per fish, a high percentage of the ciliates remained alive in the peritoneal cavity. However, with an inoculum dose of 10^5^ parasites per fish, all the ciliates died in the peritoneal cavity. Both concentrations of parasites were used in the present study. The higher concentration (10^7^ ciliates per fish) was used to investigate the gene expression in the parasites and in turbot peritoneal cells at 0, 1, 2, and 4 post-infection (hpi). The lower concentration was used to analyse the gene expression in turbot peritoneal cells at 12 and 48 hpi, comparing the response between fish injected with ciliates and fish injected with PBS. This ciliate concentration was selected because most fish cells would be engulfed and killed at 12 and 48 hpi if the higher concentrations were used.

Two parallel experiments were carried out to evaluate the changes in gene expression in P. dicentrarchi and turbot peritoneal cells during experimental infection. In one experiment, 36 fish were each injected intraperitoneally with 1 × 10^7^ ciliates (3 replicates of three fish each per time point were used; 0, 1, 2 and 4 hpi) (Appendix A). Ciliates and peritoneal cells were obtained by washing the peritoneal cavity with 5 mL of PBS. The ciliates and peritoneal cells were then counted in a Neubauer counting chamber and concentrated by centrifugation (400× *g*, 10 min). The pellet thus obtained was then diluted in 1 mL of RNAlater solution and incubated overnight at 4 oC. The samples were diluted in cold PBS (50%) and concentrated by centrifugation at 12,000× *g* for 10 min. The pellets were then frozen in liquid nitrogen and sent on dry ice to Future Genomics Technologies (Leiden, The Netherlands) for analysis.

In the second experiment, 42 fish were injected with 1 × 10^5^ ciliates and 42 fish were injected with the corresponding volume of PBS (as controls) (Appendix A). Three replicates of 7 fish each were analysed at each time point (12 and 48 hpi). The peritoneal cells were obtained and processed as described above.

In addition, a parallel experiment with the same groups of fish (3 fish per sampling time) and the same concentration of ciliates described above was used to validate the RNAseq.

### 2.4. RNA Extraction and RNAseq Analysis

Total RNA was extracted from samples preserved in RNA*later*, by using the TissueRuptor homogenizer (Qiagen Iberia, Madrid, Spain) and the miRNeasy mini kit (Qiagen Iberia, Madrid, Spain) according to the manufacturer’s instructions. RNA was eluted in RNAse-free water and quantified on a total RNA 6000 Nano series II chip (Agilent, Santa Clara, CA, USA) in an Agilent Bioanalyzer 2100 device. Illumina RNAseq libraries (150–750 bp inserts) were prepared, from 2 mg total RNA, using the TruSeq stranded mRNA library prep kit according to the manufacturer’s instructions (Illumina Inc., San Diego, CA, USA). The quality of the RNAseq libraries was checked using a DNA 1000 chip (Agilent, Santa Clara, CA, USA) and a Bioanalyzer (Agilent 2100, Santa Clara, CA, USA). RNAseq libraries were sequenced on an Illumina HiSeq2500 sequencer, as 2 × 150 nucleotides paired-end reads, according to the manufacturer’s protocol. Image analysis and base calling were performed using the standard Illumina pipeline. The read sequences were deposited in the Sequence Read Archive (SRA) (http://www.ncbi.nlm.nih.gov/sra); BioProject accession number: PRJNA648859.

### 2.5. Raw Data Cleaning, De Novo Assembly, and Gene Annotation

CLC Genomics Workbench v. 10.0.1 (CLC Bio, Aarhus, Denmark) was used for filtration and assembly and to perform the RNAseq and statistical analyses. Prior to assembly, the raw data from each sample were trimmed to remove adapter sequences and low-quality reads (quality score limit 0.05). One de novo assembly was conducted for each challenge experiment (1 × 10^7^ ciliates/fish or 1 × 10^5^ ciliates / fish). Although the turbot genome is available [28], a de novo assembly strategy was applied to analyse the *P. dicentrarchi* transcriptome. Therefore, for each experiment, all the high-quality reads were assembled in a unique file using default parameters (mismatch cost = 2, insert cost = 3, minimum contig length = 200 bp, and similarity = 0.8). The contigs yielded by these assemblies were annotated with the Blast2GO program (https://www.blast2go.com/) against the UniProtKB/SwissProt database (http://UniProt.org), applying a cut-off E-value of 1 × 10^−3^.

### 2.6. RNAseq and Differential Gene Expression Analysis

The transcriptome database generated for each experiment was used as a reference for the RNAseq analysis. Expression levels were calculated as transcripts per million (TPM) values. Finally, a differential expression analysis test (a Robinson and Smyth’s Exact Test, which assumes a Negative Binomial distribution of the data and considers the overdispersion caused by biological variability) was used to identify the DE genes. In the first experiment, the parasites at time 0 h were considered control samples, and the following comparisons were conducted: 1 hpi vs. 0 hpi, 2 hpi vs. 0 hpi and 4 hpi vs. 0 hpi. In the second experiment, the samples obtained from the infected fish were compared with the samples from the turbot injected with PBS (controls) at the corresponding sampling points. Contigs showing a >2 fold change in the absolute value relative to the control group and a Bonferroni corrected *p*-value < 0.05 were considered DE genes.

The first experiment included a mixture of contigs belonging to the parasite and contigs belonging to turbot. Two strategies were used to differentiate the contigs: (1) the DE contigs were mapped against the turbot genome [28] and those contigs with a high identity value (above 85%) were considered of piscine origin; (2) the remaining contigs were analysed by multiple BLASTx alignment (https://blast.ncbi.nlm.nih.gov/Blast.cgi) and were identified as *P. dicentrarchi* or turbot contigs on the basis of the number of hits for the preferred annotation (invertebrates vs. vertebrates).

### 2.7. Gene Ontology (GO) Assignment and Enrichment Analysis

The GO term assignments of the contig lists were obtained from the Uniprot/Swissprot BLASTx results, with Blast2GO software [29]. GO categorization of the *P. dicentrarchi* DE contigs was conducted on the basis of the biological process terms at level 2. For enrichment analysis of the turbot DE contigs (for biological processes), a Fisher’s exact test was run with default values and an FDR cut-off of 0.05 was applied.

### 2.8. RNAseq Validation by qPCR

Transcript abundance in the samples was estimated by reverse transcription-quantitative PCR (qPCR). RNA was extracted from fish tissue with TRI Reagent^®^ (Sigma-Aldrich, Madrid, Spain), according to the manufacturer’s instructions. The resulting RNA was dried, dissolved in RNase-free water and quantified in a NanoDrop ND-1000 (Thermo Fisher Scientific Inc., Wilmington, DE, USA) spectrophotometer. To prevent genomic DNA contamination, the total RNA was treated with DNAase I (Thermo Scientific, Surrey, UK). cDNA was synthesized using the cDNA synthesis kit (NZYTech, Lisboa, Portugal), with 1 μg of sample RNA. cDNA was amplified with a qPCR reaction mixture (NZYTech, Lisboa, Portugal) and 0.3 μM of each specific primer (Appendix A). For each gene, primer efficiency was determined as indicated by [30]; the expression of each gene was determined in each fish, in triplicate. A total of 5 genes in *P. dicentrarchi* (*cathepsin B*, *leishmanolysin 1589*, *leishmanolysin 17670*, *ABC transporter G family member 10* and *ABC transporter G family member 11*) and 5 genes in turbot (*tumor necrosis factor alpha*, *cd11b*, *stat6*, *c-x-c motif chemokine ligand 8* and *interleukin 1 beta*) were used to validate the RNAseq. In *P. dicentrarchi* and turbot, respectively, two genes (*β-actin* and *elongation factor 1-alpha* (*ef1α*)) and three genes (*β-actin*, *glyceraldehyde-3-phosphate dehydrogenase* and *ef1α*) were tested as candidate housekeeping genes. *Ef1*-α was identified as the most stable in both cases and was therefore selected as the reference gene for qPCR analysis. The qPCR analysis was carried out as previously described [31]. Relative gene expression was quantified by the 2^−ΔΔC^t method [32] applied with software conforming to minimum information for publication of qRT-PCR experiments (MIQE) guidelines [33]. The results are presented as the normalized expression in experimental groups, divided by the normalized expression in the control group. The concordance between RNAseq and qPCR data was evaluated using Spearman’s correlation coefficient and the null hypothesis of no differences between both methods was checked by a Wilcoxon–Mann–Whitney *U* test. The critical value was adjusted to *p*  ≤  0.05, as described by [30].

### 2.9. Statistics

The values shown in the figures are means ± standard deviation (SD). Significant differences (*p* ≤ 0.05) were determined by analysis of variance (ANOVA) followed by Tukey–Kramer multiple comparisons test.

## 3. Results

### 3.1. RNAseq Analysis and Validation

In the first experiment, in which the fish were injected with 10^7^ ciliates, about 2–2.5 × 10^6^ ciliates were recovered from each fish at all sampling times. The number of free peritoneal cells was between 2 and 3 × 10^5^ cells per fish at 0, 1 and 2 hpi and was slightly higher at 4 hpi (5 × 10^5^ cells) (Appendix A). At the time points considered (1, 2 and 4 hpi), the ciliates contained numerous engulfed peritoneal cells, particularly at 4 hpi (Appendix A). In the second experiment, no live ciliates were found in the peritoneal cavity of turbot at the sampling times (12 and 48 hpi).

Two of the replicates, corresponding to “ciliates 4 h, replicate 1” and “ciliates 4 h, replicate 2” did not produce sufficient Illumina RNAseq library for a proper run. These samples were therefore excluded from further analysis. A total of 357.3 million reads were obtained from the 22 libraries, and the number of reads obtained per sample was between 11.8 and 22.1 million, with a mean value of 16.2 million reads per library. Samples obtained at 0, 1, 2 and 4 hpi were assembled and analysed separately from samples obtained at 12 and 48 hpi. In the former, 99,821 contigs with an N50 of ~1 kb and an average length of 691 bp were obtained, and of these, 34,034 (34.1%) contigs were successfully annotated. The numbers of differentially expressed (DE) contigs (Bonferroni ≤ 0.05; FC ≥ 2), relative to the 0 h samples, were 3101, 4376 and 3499 at 1, 2 and 4 hpi respectively. About 65% of the DE contigs were annotated, and 407, 769 and 507 of these contigs were DE in *P. dicentrarchi* at 1, 2, 4 hpi (Appendix A) and the others (1627, 2067 and 1822) were DE in turbot peritoneal cells (Appendix A), respectively. In the samples obtained at 12 and 48 hpi, in which fish injected with ciliates were compared with fish injected with PBS, 167,321 contigs with an N50 of ~1.1 kb and an average length of 740 bp were obtained; of these, 31,238 (18.87%) contigs were successfully annotated. The numbers of DE contigs (Bonferroni ≤ 0.05; FC ≥ 2) in peritoneal cells of fish injected with *P. dicentrarchi*, relative to fish injected with PBS, were 464 (370 up-regulated) at 12 hpi and 57 (15 up-regulated) at 48 hpi (Appendix A). In this experiment, about 61% of the DE contigs were annotated against the UniProtKB/SwissProt database.

The expression of five genes in *P. dicentrarchi* and five genes in turbot peritoneal cells was evaluated by qPCR (Appendix A), to validate the findings of the RNAseq experiment. The RNAseq and qPCR values were strongly correlated for *P. dicentrarchi* (Spearman’s correlation ρ = 0.885) and turbot peritoneal cells (Spearman’s correlation ρ = 0.846).

### 3.2. Changes in Gene Expression in P. dicentrarchi during an Experimental Infection

In order to identify which parasite genes may be important during the early stages of infection and which may play a role in virulence or in resistance against the fish immune response, the changes in gene expression in the scuticociliate were analysed during infection. Differential expression of 407 (369 up-regulated (UR)), 769 (415 UR) and 507 (119 UR) genes was observed at 1, 2 and 4 hpi, respectively.

GO analysis was carried out to explore the biological processes that were best represented during infection (Figure 1). Five categories were well represented at 1 and 2 hpi, including biosynthetic processes, catabolic processes, biogenesis, proteolysis and transmembrane transport, with a similar distribution at both times. At 4 hpi, there was a substantial increase in the percentage of genes involved in catabolic processes and proteolysis (Figure 1).

The most strongly regulated genes in *P. dicentrachi* during infection were those related to the ATP-Binding Cassette (ABC) transporter gene family, which included 12 DE genes (Table 1). Of these, *ABC transporter G family member 10* (*abcg10*) was strongly UR at 1, 2 and 4 hpi. Other genes that were UR at the three sampling times were *abca4* and *abcb4*. Six genes were DE only at 1 and 2 hpi, including three members of the G family, with *abcg14* and *abcg11* being strongly UR. Overall, the highest expression of ABC transporter family genes was found at 1 hpi. Finally, *abca3* and *abcb1* were down-regulated (DR) at 4 or at 2 and 4 hpi, respectively (Table 1).

Several leishmanolysin-like genes were also DE. Four genes of that family were UR and two genes were DR (Table 1). As *P. dicentrarchi* proteases are considered virulence factors, we were interested in determining how the genes of lysosomal enzymes would behave during the early stages of infection. The expression of several cathepsins, especially *ctsb, ctsl* and *ctsd*, was DR at 2 and 4 hpi (Table 1). In addition, several other genes that coded for lysosomal proteins, such as *lysosomal acid phosphatase* and *lysosomal alpha-mannosidase* were also DR at 2 and 4 hpi (Table 1).

Tubulins and tubulin associated proteins, such as kinesins and dyneins, play key roles in, e.g., ciliate division and movement. Apart from *kinesin family member 1*, expression of alpha and beta tubulins and many kinesins and dyneins (including the *outer dynein arm 1* gene, which is required for ciliary motion) was DR at 2 and 4 hpi (Table 1).

Ribosome synthesis plays a central role in regulating cell growth [34]. Many genes involved in ribosome biogenesis, DNA transcription and RNA translation were UR (Table 2). The strongest regulation occurred in the gene *regulator of rDNA transcription 15*. For most genes, UR occurred at 1, 2, and 4 hpi, peaking at 2 hpi (Table 2).

Analysis of the genes involved in the cell cycle showed that the *cyclin-dependent kinase 1* (*cdk1*) gene, which is involved in mitotic control, was DR. Similarly, *cyclin-B2-3* and *cyclin-dependent kinase regulatory subunit 2*, which participate in the cell cycle, were also DR. By contrast, *cyclin-dependent kinase 2* (*cdk2*), which participates in G1-S transition and the DNA damage response, was strongly UR at 1, 2 and 4 hpi (Table 2).

Genes involved in glycolysis, such as *triose-phosphate isomerase*, or in the Krebs cycle, such as *isocitrate dehydrogenase*, *citrate synthase* and *acetyl-coenzyme A synthetase*, and components of the electron transport chain, such as mitochondrial *cytochrome c*, were DR at 2 and 4 hpi. Other genes involved in the mitochondrial electron transport chain, such as *cytochrome c oxidase subunit 1* and *2*, were also DR at 2 and 1 hpi, respectively (Table 2). Finally, genes coding for enzymes of the microsomal compartment possibly involved in detoxification processes were also DR, including several cytochrome p450 (*cyp3a19*, *cyp4b1*, *cyp4e3*) and *glutathione S-transferase 2* and *3* genes (Table 2).

### 3.3. P. dicentrarchi Generates an Intense Inflammatory Response in Turbot during the Aarly Stages of Infection

We observed intense regulation of many genes of peritoneal cells at 1, 2 and 4 hpi. The response was much lower at 12 hpi and almost disappeared completely at 48 hpi in those fish that were able to kill *P. dicentrarchi* during the first few hours after injection. To analyse their function, DE genes were subjected to gene ontology and enrichment analysis for biological processes, molecular function and cellular components. The proportion of genes involved in different biological processes was very similar across groups at 1, 2 and 4 hpi (Figure 2). DEGs at 1, 2 and 4 hpi were associated with the apoptotic process (between 7.9 and 8.4%), inflammatory response (between 4.3 and 4.8%), immune response (3.4 to 5.8%), Fc-epsilon receptor signalling pathway (4.3 to 4.6%), innate immune response (3.5 to 3.6%), cell adhesion (3.3 to 3.8%), phagocytosis (2.8 to 3.1%) and the cell surface receptor signalling pathway (2.5–2.7%) (Figure 2). Many genes of signalling pathways or cascades were also DE, such as those involved in the MAPK cascade (3.6 to 3.8%), the NIK/NF-kappaB signalling pathway (1.7 to 1.9%) and the I-kappaB kinase/NF-kappaB signalling pathway (1 to 1.4%). However, the proportion of genes involved in the defence response to bacterium (2.3 to 2.9%) was higher than the number of genes involved in the defence response to protozoan (0.5 to 0.6%) (Figure 2).

Injection of ciliates into the peritoneal cavity generated a very potent inflammatory response in turbot, with the peritoneal cells expressing genes related to this response, even though many of them were already phagocytosed by the ciliate. One of the first events that occur during the host response to pathogens is recognition by host pattern recognition receptors. In peritoneal cells, *toll-like receptor* (*tlr*) *2* was UR at 1 and 2 hpi, *tlr13* was UR at 1, 2, 4 and 12 hpi, and *tlr5* was UR at 2, 4 and 12 hpi. Of these three genes, *tlr5* was the most strongly regulated (Table 3). Adaptor *myd88*, which is involved in almost all TLRs signalling pathways, was UR. Other genes that were DE included *interleukin-1 receptor-associated kinase 4* (*irak4*), several other components of the NF-κB pathway, including *nuclear factor nf-kappa-b p100 subunit* (*nfkb2*) or *nf-kappa-b inhibitor alpha* (*nfkbia*) or *map kinase kinase kinase 14* (*map3k14*). In addition, several components of the MAPK and JAK/STAT pathways were also DE (Table 3). Except for *mapk6*, which was also UR at 12 hpi, the other components of the MAPK pathway were differentially expressed at 1, 2 and 4 hpi, and most of them peaked at 4 hpi. In the genes related to the JAK/STAT pathway, *stat1, stat3, stat4* and *stat6* or *jak1* and *jak3* were UR at 1, 2 and 4 hpi, peaking at 4 hpi. However, negative regulators of the pathway, *socs1* and *socs3*, were also UR at 1, 2, 4 and 12 hpi (Table 3).

During infection by *P. dicentrarchi*, several chemokines were DE, including *c-x-c motif chemokine ligand 8* (*cxcl8*) and *c-c motif chemokines* (*ccl*) *11*, *3*, *4*, *2* and *20*. *Cxcl8* was strongly expressed at 1, 2 and 4 hpi, but was DR at 48 hpi, and *ccl20* was only UR at 2 hpi. In addition, several chemokine receptors were also UR, including *c-x-c chemokine receptor type 4*, *1*, *3*, *3-2*, and *2* and *chemokine-like receptor 1* (*cmklr1*). *Cxcr4* was most strongly expressed at 1, 2 and 4 hpi, and *cxcr1* and *cmklr1* were UR at 1, 2, 4 and 12 hpi (Table 4).

The genes *prostaglandin g h synthase 2* and *prostaglandin e synthase 3*, which code for enzymes involved in the synthesis of prostaglandin E2, were up-regulated, the former at 1, 2 and 4 hpi, and the latter at 1 hpi (Table 4). Genes coding for enzymes involved in the synthesis of leukotrienes, i.e., *12s-lipoxygenase* and *leukotriene a-4 hydrolase*, were UR at 1, 2 and 4 hpi. Positive differential expression of prostaglandin and leukotriene receptors, i.e., *leukotriene b4 receptor 1* and *prostaglandin i2 receptor*, was also observed. Finally, *arachidonate 5-lipoxygenase* was only UR at 12 and 48 hpi and *arachidonate 15-lipoxygenase* only at 48 hpi (Table 4).

Several interleukins (IL) and IL receptors were also DE during *P. dicentrarchi* infection (Table 5). For most of these, the response peaked at 4 hpi. The pro-inflammatory *il1β* was strongly UR at 1, 2 and 4 hpi and, similarly to *cxcl8*, was DR at 48 hpi. Other interleukins that were DE at 1, 2 and 4 hpi included *il16*, *il27β* and *myeloid-derived growth factor* (also *il27*). However, *il10* was only UR at 12 hpi and *il12β* was UR at 12 and 48 hpi. Numerous IL receptors were also UR, including *il6ra*, *il6rb*, *il22r2*, *il1r1*, *il1r2*, *il10r1*, *il3b2*, *il7ra*, *il2rb*, *il2rg* and *il31r*. Of these, *il6ra*, *il22r2* and *il1r2* were most strongly regulated. In addition, *il1r1*, *il1r2*, *il10r1* and *il3b2* were DE at 12 hpi (Table 5).

Among all the cytokines, up-regulation was strongest in the pro-inflammatory cytokine *tumour necrosis factor-alpha* (*tnfa*) at 2 and 4 hpi, peaking at 4 hpi. Many other genes related to TNF, including TNF receptors, were also differentially expressed, including *tnf alpha-induced protein 8 like 2*, *tnf ligand superfamily member 6*, *tnf superfamily member 13b*, *tnf receptor superfamily member 26*, *tnf alpha-induced 2, tnf receptor superfamily member 1b*, *tnf receptor-associated factor 2*, *tnf receptor superfamily member 5* and *tnf receptor superfamily member 11b*. In almost all cases, expression of TNF and TNF-related genes peaked at 4 hpi (Table 5).

Another group of genes that were DE, but at a much lower level than the previously mentioned genes, was the group of interferon (IFN) related genes, including several interferon regulatory factors and other IFN related genes. Most of these were up-regulated at 1, 2 and 4 hpi, but some of them were also up-regulated 12 h after intraperitoneal injection. Among this group of genes, *interferon-double-stranded rna-activated kinase* and *interferon-induced 44-like* were the most strongly regulated (Table 6).

Several genes associated with cell death were DE at 1, 2 and 4 hpi, but many were also DE at 12 hpi (Table 7). Some members of the *bcl-2 gene family*, which are regulators of cell apoptosis, were strongly UR at 1, 2 and 4 hpi, including *apoptosis facilitator bcl-2 14*, *bcl2 associated agonist of cell death*, *apoptosis regulator bax*. Other DE genes were involved in processes that occurred during apoptosis, such as DNA fragmentation, including *dna fragmentation factor subunit beta*, or the extrinsic pathway of apoptosis, including *fas cell surface death receptor* and *fas-associated death domain*, which are associated with Fas ligand cell death. We also observed up-regulation of *caspases*, such as *caspase 3* and *8*, which were UR at 2 and 4 hpi or at 12 hpi, respectively (Table 7). Finally, several apoptotic regulator genes were also differentially expressed, including *programmed cell death 1 ligand 1* and *programmed cell death 4*, *6* and *10*.

Several genes related to T and B lymphocytes or other immune cells were DE (Table 8). Up-regulation of components of IgM and IgD was observed at 1, 2 and 4 hpi, such as *immunoglobulin heavy constant mu* and *immunoglobulin delta heavy chain*, and several genes involved in B cell development and function in mammals, including *b-cell antigen receptor complex-associated alpha chain* and *b-cell receptor cd22*. Several genes associated with cytotoxic T lymphocytes and natural killer cells, such as *granzyme a*, *granzyme b* and *perforin-1*, were also UR. (Table 8). In addition, genes that are expressed by activated B and T lymphocytes in mammals, but that are also markers of dendritic cells, e.g., *cd83 antigen*, were also UR (Table 8). Finally, genes that can be expressed in mast cells, such as *high-affinity immunoglobulin epsilon receptor subunit gamma* and *mast cell protease 1a*, were DE at 1, 2 and 4 hpi.

Several genes related to the complement and the coagulation systems were DE. Some components of the complement system, including *complement c1q subcomponent subunit b, complement c1s subcomponent, complement factor d, complement factor h, c3* and *c4* were UR at 1, 2 or 4 hpi, depending on the gene. Similarly, some components of the coagulation system were also DE at 1, 2 and 4 hpi, including *coagulation factor VIII*, *coagulation factor XIII a chain* and *tissue factor pathway inhibitor*. However, *tissue factor* was only UR at 12 hpi (Table 9).

## 4. Discussion

In the present study, we evaluated gene expression in the ciliate parasite *P. dicentrarchi* at 1, 2 and 4 hpi, relative to the gene expression in samples processed at 0 h, during experimental infection in the turbot *S. maximus*, to identify the genes participating in this process, including those potentially involved in virulence or in resistance to the host immune system. The most strongly regulated genes in *P. dicentrarchi* during infection were the ABC transporters, which were mainly UR. Protozoan ABC transporters are involved in nutrient transport but they also protect cells from both internally produced and exogenous toxins, and some have been associated with antiparasitic drug resistance as well virulence and oxidative stress [35,36,37]. These transporters are very abundant in ciliates, and 165 ABC transporter genes were identified in the macronuclear genome of the ciliate *Tetrahymena thermophila* [38]. In the present study, the *P. dicentrarchi* ABC transporter G family genes showed a strong, early response during infection, indicating that some ABC transporters are very important in this process. It is not known whether they are involved in ciliate resistance against the attack of fish immune system or whether they have other roles in infection.

Very few free-living ciliate species can invade the fish, resist attack by the immune system and cause systemic infection. However, *P. dicentrarchi* is capable of infecting cultured fish and causing massive mortalities. Thus, the ciliates must be able to resist the attack from the fish immune system and also to destroy fish tissue, for which proteases may be important. In the present study, we observed an increase in the expression of several genes of the leishmanolysin family, which are membrane-bound metalloproteases capable of degrading and cleaving many biological molecules [39]. Leishmanolysin GP63 is a major surface protein and is considered one of the main virulence factors in the human pathogen *Leishmania* [40]. GP63 degrades a large number of proteins, including complement [41], and down-regulation of gene expression makes the parasites more susceptible to complement-mediated lysis [42]. The complement system is considered important in defence against *P. dicentrarchi* [11,20], and although complement levels are lowered in the serum of infected turbot [12], it is not known whether leishmanolysins participate in the process. Our results agree with those obtained in previous studies, which reported high level of leishmanolysin expression in a *P. dicentrarchi* isolate (I1) obtained from turbot infections [43], and UR of several leishmanolysin genes in *Miamiensis avidus* fed a cell line [44], suggesting that these molecules play a role in cell degradation. Thus, although the role of each regulated leishmanolysin gene in the success of *P. dicentrachi* infection remains to be established, it appears that leishmanolysins are probably involved in ciliate virulence.

Ciliate proteases, other than leishmanolysins, are considered virulence factors in *P. dicentrarchi* [10,12,13,14]. However, several members of the cathepsin family were DR during the early stages of *P. dicentrarchi* infection. Interestingly, other genes whose proteins are also located in the lysosome were also DR. Because some ciliates contained phagocytosed turbot cells, release of lysosomal enzymes probably occurred during the early stages of infection. However, the expression of several cathepsin genes also did not increase in the scuticociliate *Anophryoides haemophila* during an infection in American lobster [45]. Although the expression of these genes will probably be enhanced at later stages in order to degrade the endocytosed material, the expression of these enzymes may be delayed during infection due to other priorities of the cells.

Many genes associated with rDNA transcription, ribosome biogenesis or rRNA synthesis were UR in *P. dicentrarchi* during the early stages of infection, indicating a focus on ribosome production and protein synthesis in the ciliate. However, except for *kinesin family member 1*, the *tubulin alpha* and *beta* genes and genes coding for several dyneins and kinesins were DR in *P. dicentrarchi* during infection at 2 and/or 4 hpi. Tubulins and the associated motor proteins play many roles in eukaryotic cells, and particularly in ciliates, including cell division or cell movement. Genes involved in cell division, such as *cyclin-dependent kinase 1* (*cdk1*) and *cyclin B*, were also DR in *P. dicentrarchi* during infection. *Cdk1* is a key protein that regulates mitotic entry and spindle assembly and its activation depends on *cyclin B* [46]. Our findings suggest that cell division is not activated during the early stages of infection. However, other genes associated with the cell cycle, such as *cdk2*, were strongly UR. *Cdk2* activation leads to DNA replication in the cell, and it is also involved in DNA damage and the DNA repair response [47]. Some ciliates probably suffered DNA damage because of attack from the turbot immune system, thus explaining the increase in *cdk2* expression. Several genes involved in glycolysis, the tricarboxylic acid (TCA) cycle and the mitochondrial respiratory chain were DR at 2 and (particularly) 4 hpi. If the *P. dicentrarchi* mitochondrial genome is similar to that of *Tetrahymena pyriformis* and *Paramecium aurelia* [48], the genes of the respiratory chain will be located in the mitochondrial genome, while those of the TCA cycle and glycolysis will be located in the nucleus. Unfortunately, little is known about how the expression of these genes is regulated in ciliates during infection. Many ciliate mitochondria may be altered because of interactions between the ciliate and the fish immune system. Previous studies have shown that turbot phagocytes are strongly stimulated by contact with *P. dicentrarchi*, releasing a high amount of reactive oxygen species (ROS) [11]. Leucocytes do not seem to produce sufficiently high concentrations of toxic substances to kill the parasite, but some ciliate components may be affected, as an increase in ROS levels is associated with alterations in the *P. dicentrachi* mitochondria [49]. Due to retrograde regulation, mitochondrial dysfunctions may affect the expression of enzymes involved in the TCA cycle, as observed in cells in other organisms [50], thus explaining the DR of these genes.

The present study shows that intraperitoneal injection of turbot with *P. dicentrarchi* induced an acute inflammatory response and enhanced the expression of genes involved in inflammatory pathways, inflammatory cytokines or genes coding for enzymes involved in the synthesis of prostaglandins and leukotrienes. Recruitment of inflammatory cells, which is part of the inflammatory response [51], was not evaluated in the present study but has been demonstrated in previous studies [21]. Recognition of the pathogen by pattern recognition receptors is an important event in the inflammatory process. High levels of expression of *tlr5* and, to a lesser extent, of *tlr13* and *tlr2*, occurred in turbot peritoneal cells although *tlr2* and *tlr13* expression began earlier. *Tlr2* expression increased in some organs of channel catfish (*Ictalurus punctatus*) and orange-spotted grouper (*Epinephelus coioides*) during infection with the ciliate ectoparasites *I. multifiliis* and *C. irritans* respectively [24,52,53]. Studies in mammals have shown that glycosylphosphatidylinositol (GPI) is a potent activator of TLR2 in several protozoan parasites and that GPI is recognised by this receptor [54,55]. It has also been suggested that TLR2 mediates activation of the MAPK and NF-κB pathways [56]. An increase in the expression of genes involved in these pathways was also observed in the present study, indicating that turbot TLR2 is activated during *P. dicentrarchi* infection and that it may be important in the recognition of some ciliate molecules. However, because *P. dicentrarchi* causes extensive damage in tissues and TLR2 can also be activated by several damage-associated molecular patterns [57], an increase in the expression of this receptor associated with any of these molecules cannot be ruled out. Expression of *tlr13* and particularly *tlr5* was strongly enhanced in turbot peritoneal cells, peaking at 4 hpi and lasting until at least 12 hpi. TLR13 is located in endolysosomes and recognises bacterial 23s ribosomal RNA in mice, while TLR5 is located at the plasma membrane and recognises bacterial flagellin [58]. Although TRL13 has been associated with the resistance of some insects to parasites [59], it is not known whether it plays a role in fish parasite infections. Fish *tlr5* was also strongly up-regulated in some tissues of *E. coioides* and in the Tibetan highland fish (*Gymnocypris przewalskii*) in response to infection with the ciliates *C. irritans* [24,60] and *I. multifiliis* [61] respectively, although, as far as we know, no particular role for this receptor in parasite recognition has been reported. The UR of *tlr5* found in the present study may be a consequence of prior stimulation of other cell receptors. In this respect, we cannot rule out unspecific stimulation of TLRs generated by a substance released by the parasite or even by a mechanical effect generated by the contact between the parasite membrane (which is moving all the time due to the movement of the cilia) and the leukocyte membranes.

Regarding the stimulation of inflammatory pathways, many components of NF-κB, MAPK and JAK/STAT pathways were UR. Some of those genes, such as *myd88*, which was UR at 1, 2 and 4 hpi, have a positive effect on the pathways; however, others, such as *nfkbia, socs1* and *socs3,* which were UR at 1, 2, 4 and 12 hpi, exert a negative effect in order to limit the inflammatory response [62]. NF-κB, MAPK and JAK/STAT are involved in inflammatory processes, inducing expression of genes that regulate the inflammatory response, although they can also participate in other processes [63,64,65]. Several chemokines and their receptors were UR in turbot peritoneal cells during *P. dicentrarchi* infection. Among these, the strongest regulation was found in *cxcl8* and their receptors *cxcr1* and *cxcr2*, which can be expressed by several types of leukocytes, especially neutrophils; *cxcl8* expression has been associated with neutrophil recruitment [66]. This chemokine also induces chemotaxis of neutrophils and, to a lesser extent, lymphocytes and macrophages in turbot [24,67]. These results support those demonstrating intense migration of neutrophils to the turbot peritoneal cavity during *P. dicentrarchi* infection [21]. Several interleukins and their receptors, as well as several members of the TNF family and their receptors, were also strongly UR in turbot peritoneal cells, particularly the proinflammatory cytokines IL1b and TNFa. These and other inflammatory cytokines are produced by leukocytes, due to TLR stimulation, and promote inflammation [58]. In addition, molecules such as TNFa can modulate multiple signalling pathways, some of which are related to the immune response during inflammation and also other responses, such as cell death [68]. The role of both IL1b and TNFa in regulating the inflammatory response is well conserved in fish [69]. Several other interleukins, interleukin receptors and members of the TNF superfamily that participate in the regulation of the inflammatory response are also strongly expressed in turbot peritoneal cells, including *il27b, il6ra, il6rb, il22r2, il3b2, il3b2, tnfaip8l2b, tnfaip2* and others, some of which, such as *tnfaip8l2b*, are negative regulators of the immune response [70], indicating the complexity of the response generated. Some of the interleukin genes such as *il10* and *il10r* were UR at 12 hpi and at 1, 2, 4 and 12 hpi, respectively. IL10 displays potent anti-inflammatory and regulatory activity in most immune processes during infection [71], suggesting an anti-inflammatory state in fish capable of controlling the infection. Increased expression of *il12b* was observed at 12 and 48 hpi, although to a lower extent than for *il10*. However, there was no increase in the expression of the complementary subunit or the receptors. In mammals, IL12B is secreted by phagocytic cells and acts on T and NK cells, inducing IFN-γ production and generating a proinflammatory state [72]. An increase in IFN*-γ* has been observed in leukocytes of other flatfish species stimulated with recombinant Il12 [73]. However, *ifn-γ* expression did not increase in turbot peritoneal leukocytes in the present study, suggesting that in this case, an increase in *il12b* expression will not generate a proinflammatory state.

Several genes of the interferon family were UR, although to a much lesser extent than those related to other cytokines. *Interferon-induced double-stranded RNA-activated kinase* was one of the genes in which DE was highest, being up-regulated at 1, 2, 4 and 12 hpi. This gene is a key component of host innate immunity that restricts viral replication and propagation but also participates in the stress response, being crucial for cell survival and proliferation, functions that are beyond the viral response [74]. Most of the other DE genes were interferon regulatory factors (IRFs). Some of these factors are involved in TLR signalling pathways, inducing the expression of *type 1 ifn* or proinflammatory cytokine genes [75]. IRFs have been shown to play several roles in parasite infection, particularly *irf8*, which appeared UR at 1, 2, 4 and 12 hpi in turbot peritoneal cells during *P. dicentrarchi* infection and which regulates the production of proinflammatory cytokines during malarial infections [76]. However, IRFs are also involved in differentiation and apoptosis of immune cells in mammals [75] and in fish [77]. All of these events seem to occur in the turbot peritoneal cavity, and different IRFs may participate in any of them.

Many genes associated with apoptosis were highly UR in turbot peritoneal cells during initial infection with *P. dicentrachi*. Regulation of most genes occurred at 1, 2, 4 hpi, peaking at 4 hpi, as well as at 12 hpi in some. Previous studies have shown that *P. dicentrarchi* proteases induce apoptosis in turbot cells [14]. However, increased expression of genes related to apoptosis or in the number of apoptotic cells in turbot peritoneal cells after injection with other stimuli have also been observed [31,78]. Other studies have also shown that large numbers of neutrophils migrate to the peritoneal cavity after injection of *P. dicentrarchi* [21] and many of these cells seem to initiate the expression of genes leading to apoptosis. In later stages, these cells can be phagocytosed by macrophages, contributing to the resolution of inflammation [79]. On the basis of these findings, it appears that most of the gene expression associated with apoptosis in turbot peritoneal cells occurs in neutrophils. However, it has been shown that other stimuli also cause apoptosis in fish peritoneal macrophages [80], indicating that other cell types may also be involved in this process.

Increased expression of several T and B lymphocyte related genes was also observed in the turbot peritoneal cells. Thus, e.g., *igm* and *igd heavy chains*, *cd79a* and *cd22*, which are expressed in B lymphocytes, were UR. This regulation may be related to the recruitment of B cells to the peritoneal cavity, as shown in other fish species after injection with different stimuli [81]. Several genes related to cytotoxicity were also DE. *Granzyme A* was UR at 1, 2 and 4 hpi and *granzyme b* and *perforin-1* were UR only at 1 hpi. These genes are produced by cytotoxic lymphocytes in mammals and fish [82,83]. Information about how these molecules interact with parasites in fish is scarce. However, in mammals, it has been observed that in addition to killing parasite-infected cells, these cytotoxic granule effectors can kill parasites [84]. It is not known whether these molecules can kill *P. dicentrarchi*. However, a turbot NK-lysin, an effector molecule of cytotoxic T lymphocytes and NK-cells, has been shown to be toxic for *P. dicentrarchi*, although infection did not clearly affect mRNA expression [19]. On the basis of these observations, these molecules from cytotoxic cells appear highly likely to play a role in *P. dicentrarchi* infection.

Finally, several genes related to the coagulation and the complement systems were also DE in turbot during *P. dicentrarchi* infection. Several complement genes have been found to be expressed in immune organs of turbot during a *P. dicentrarchi* infection [23], as well as locally in *E. coioides* infected with the ciliate *C. irritans* [24] and also in the skin or liver of common carp (*Cyprinus carpio*) infected with *I. multifiliis* [85]. The findings of the present study are consistent with the aforementioned findings in suggesting that peritoneal cells express complement and that the expression is regulated during infection. Similar observations have been made for genes involved in coagulation. Both the complement and the coagulation systems play important roles in defending turbot against *P. dicentrarchi* [20,21], and the complement or coagulation proteins released by leukocytes at the site of infection may represent an important additional source of these molecules.

How does the immune response generated by *P. dicentrarchi* in turbot compare with that induced by other parasites in fish? The immune response induced in fish by different parasites varies widely. Thus, some parasites do not elicit a host response, while others induce a more or less intense inflammatory response [86,87]. Previous transcriptomic studies carried out in fish infected with ciliate parasites have shown that these parasites usually generate an inflammatory response. However, the sampling times were longer than used in the present study, and the responses are therefore not necessarily comparable. Transcriptomic analysis, involving a microarray enriched in immune genes, of spleen, kidney and liver of turbot infected with *P. dicentrarchi*, revealed strong up-regulation of many immune response genes [22,23] at 1 dpi, but a less intense response than observed in turbot peritoneal cavity in the present study. Sampling times after experimental infection with ciliates were 12 h [88], 24 h [25] or 3 dpi [24] in fish infected experimentally with *C. irritans*. In the ectoparasite *I. multifiliis*, sampling was conducted 2 dpi [61] or 8 dpi [27]. However, regardless of the differences between the sampling times in the present and previous studies, ectoparasite ciliates such as *C. irritants* also induced a potent inflammatory response on the skin of infected fish, with up-regulation of genes involved in innate immunity, genes coding for TLRs, TLR signalling pathways, chemokines and chemokine receptors and genes associated with complement activation [24]. Similarly, *I. multifiliis* generated a massive immune response in the gills of rainbow trout (*Oncorhynchus mykiss*) [27].

## 5. Conclusions

In conclusion, many genes of the ABC transporter and leishmanolysin gene families were strongly regulated in the ciliate parasite *P. dicentrarchi* during the early stages of infection in turbot, suggesting that they play an important role in this process. During infection, the ciliate generated a very high inflammatory response at the site of injection, with the regulation of most of the genes known to be involved in the inflammatory response in mammals, indicating that turbot use all the protective mechanisms they have available to prevent entry of the parasite. The findings also provide some valuable insight into how the acute inflammatory response occurs in fish.

## Figures and Tables

**Figure 1 biology-09-00337-f001:**
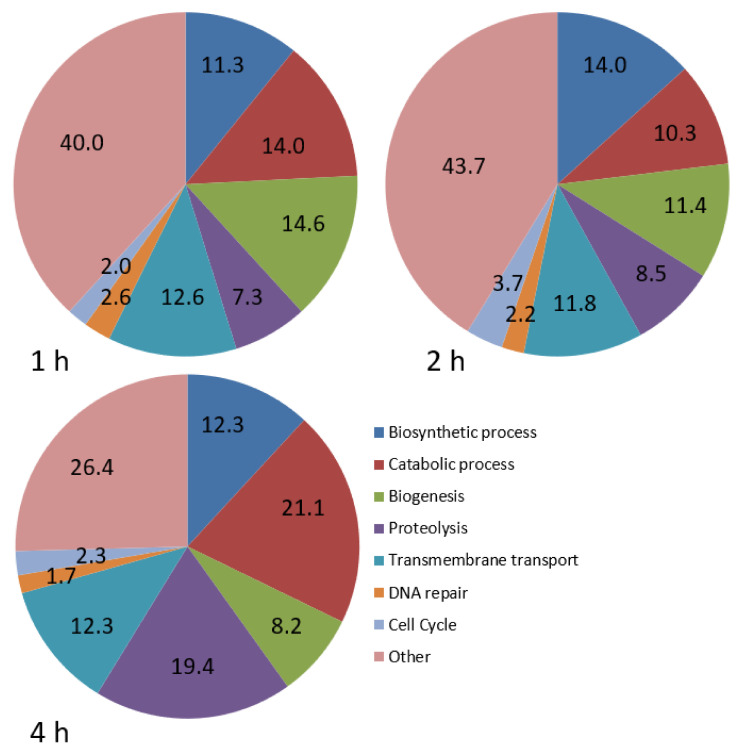
Pie charts showing the proportion (%) of differentially expressed genes involved in various biological processes in *P. dicentrarchi* at 1, 2 and 4 hpi, relative to the number at 0 h.

**Figure 2 biology-09-00337-f002:**
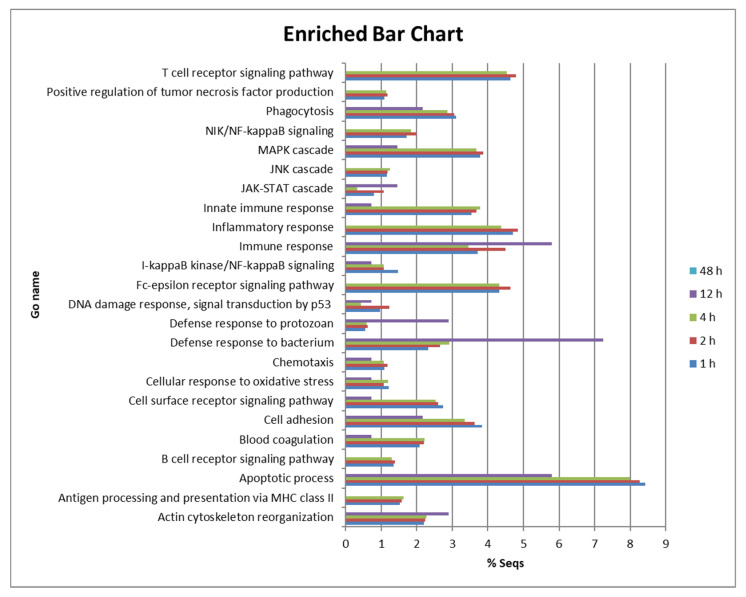
Bar chart showing the proportion (%) of DE genes involved in various biological processes in turbot peritoneal cells at 1, 2 and 4 hpi.

**Table 1 biology-09-00337-t001:** Heat map showing a group of differentially expressed (DE) genes in *P. dicentrarchi,* including ABC transporters, leishmanolysins and genes related to microtubules and to lysosomes, at 1, 2 and 4 hpi. Results are expressed as mean expression ratios of groups 1, 2 and 4 hpi vs. 0 h. Red indicates increased gene expression levels; green indicates decreased levels.

Gene Name	Gen Abbrev.	1 h	2 h	4 h
*ABC transporter G family member 10*	*abcg10*	1547.333077	1525.204333	135.9495633
*ABC transporter G family member 14*	*abcg14*	155.981555	66.32506772	
*ABC transporter G family member 11*	*abcg11*	117.2851641	31.01838328	
*ABC transporter G family member 22*	*abcg22*	9.047137204	6.428493341	
*ABC transporter F family member 3*	*abcf3*	5.149616627	5.972621581	
*ABC transporter A family member 2*	*abca2*	17.05398277	7.274235391	
*ABC transporter A family member 4*	*abca4*	17.12815109	26.00467446	39.15870736
*ABC transporter B family member 4*	*abcb4*	8.243947246	3.745339799	6.402422485
*ABC transporter A family member 7*	*abca7*	3.043246918	3.532536485	
*ABC transporter B family member 3*	*abcb3*	3.36028248		
*ABC transporter A family member 3*	*abca3*			−7.218505508
*ABC transporter B family member 1*	*abcb1*		−2.872487246	−4.36433424
*leishmanolysin 17670*	*lmln17670*	13.02222148	56.71743416	13.79572431
*leishmanolysin 4908*	*lmln4908*	4.636533774	6.397241791	12.09714031
*leishmanolysin 6891*	*lmln6891*	6.660199714	15.9960053	
*leishmanolysin 906*	*lmln906*	2.789707738	3.478134169	
*leishmanolysin 8301*	*lmln8301*	−3.341460004	−5.425240786	
*leishmanolysin 1589*	*lmln1589*	−7.499602753	−19.34465739	−29.97419775
*tubulin alpha chain*	*tba*		−3.541692478	
*tubulin beta chain*	*tbb1*		−3.494382907	
*dynein alpha flagellar outer arm*	*dyha*			−4.485994554
*dynein heavy chain axonemal heavy chain 7*	*dyh7*		−2.931765009	−5.770286234
*dynein heavy cytoplasmic*	*dyhc1*		−3.566132897	−13.96776319
*dynein light chain cytoplasmic*	*dyl2*		−2.781891083	
*dynein regulatory complex 1*	*drc1*			−4.298212013
*outer dynein arm 1*	*oda1*			−4.407810139
*kinesin 7l*	*kn7l*		−5.659610804	
*kinesin family member 1*	*kif1*	15.69290132	13.68478782	
*kinesin fla10*	*fla10*		−3.064471017	−4.729088437
*kinesin kif15*	*kif15*	−2.943783569	−6.688399519	−11.43932373
*kinesin-ii 95 kda subunit*	*krp95*		−8.465050495	
*cathepsin d*	*ctsd*	3.7	−2.45	−5.7
*cathepsin l*	*ctsl*		−4	−18.8
*cathepsin z*	*ctsz*		−3.3	
*cathepsin b*	*ctsb*		−5.3	−41.49
*lysosomal acid phosphatase*	*ppal*		−5.19988665	−11.71489074
*lysosomal alpha-mannosidase*	*mana*		−4.216965484	−9.663142631

**Table 2 biology-09-00337-t002:** Heat map showing a group of DE genes in *P. dicentrarchi,* including genes related to ribosome biogenesis, DNA transcription, cell cycle, metabolism, mitochondrial respiratory chain and detoxification, at 1, 2 and 4 hpi. Results are expressed as mean expression ratios of groups 1, 2, and 4 hpi vs. 0 h. Red indicates increased gene expression levels; green indicates decreased levels.

Gene Name	Gen Abbrev.	1 h	2 h	4 h
*regulator of rdna transcription 15*	*rrt15*	−19.2921801	305.2526145	14961.40172
*pre-rrna-processing tsr1 homolog*	*tsr1*	8.39214507	17.7662126	12.40935626
*ribosome biogenesis regulatory homolog*	*rrs1*	8.397288759	17.61314827	10.18017877
*60s ribosome subunit biogenesis nip7*	*nip7*	11.22783554	17.445352	
*ribosome biogenesis bop1*	*bop1*	7.627651039	13.73086419	5.8194392
*ribosome production factor 1*	*rpf1*	6.256019845	11.74039515	5.947145101
*u3 small nucleolar rna-associated 6*	*utp6*	7.9733096	17.52126323	7.607224123
*eukaryotic translation initiation factor 6*	*eif6*	3.03261779	4.997917367	
*eukaryotic translation initiation factor 3 subunit l*	*eif-3*	2.695718872	4.426628843	
*atp-dependent rna helicase has1*	*has1*	11.52254521	16.62575106	5.879612164
*rna polymerase i subunit 1*	*rpa1*	6.376072721	9.0470461	5.741738543
*rna polymerase i subunit 2*	*rpa2*	4.419921248	7.480794029	
*cyclin-dependent kinase 2*	*cdk2*	179.5992883	233.6336546	119.049648
*cyclin-dependent kinase 1*	*cdk1*	−3.206564057	−9.082843233	−8.784823344
*cyclin-b2-3*	*ccnb23*		−2.98467857	−5.526397608
*cyclin-dependent kinases regulatory subunit 2*	*ppp2r1a*		−4.068280735	−4.813378142
*centrosomal of 78 kda*	*cep78*		−4.499861387	−7.279573252
*triose-phosphate isomerase*	*tpi1*			−5.037605277
*isocitrate dehydrogenase*	*idhp*		−3.619776764	−18.74348161
*citrate synthase*	*cs*		−3.677561365	−12.28402905
*acetyl-coenzyme a synthetase*	*acsa*		−4.375876016	−8.810778878
*cytochrome c mitochondrial*	*ccpr*		−3.056139192	−7.538025415
*cytochrome c oxidase subunit 1*	*cox1*		−14.4260752	
*cytochrome c oxidase subunit 2*	*cox2*	−27.83893378		
*hypoxia up-regulated 1*	*hyou1*	−3.4739462	−8.629031534	−8.645030215
*cytochrome p450 3a19*	*cyp3a19*			−22.82674295
*cytochrome p450 4b1*	*cyp4b1*		−3.684326503	
*cytochrome p450 4e3*	*cyp4e3*			−28.32575971
*glutathione s-transferase 2*	*gstm2*		−5.124694687	−15.81386515
*glutathione s-transferase 3*	*gst3*		−5.175570202	−24.89756053
*glutathione s-transferase theta-2b*	*gst*		−2.786019803	
*trichocyst matrix t1-b*	*t1-b*		−13.95	−14.54
*trichocyst matrix t2-a*	*t2-a*	−3.67	−12.13	−13.88
*trichocyst matrix t4-b*	*t4-b*	−3.92	−8	−14.84

**Table 3 biology-09-00337-t003:** Heat map showing a group of DE genes in turbot peritoneal cells at 1, 2, 4 and 12 hpi, including genes coding for toll-like receptors and genes involved in different signalling pathways. Results are expressed as mean expression ratios of groups 1, 2 and 4 hpi vs. 0 h, or 12 hpi vs. phosphate-buffered saline (PBS). There were no DE genes at 48 hpi. Red indicates increased gene expression levels; green indicates decreased levels.

Gene Name	Gene Abbrev.	1 h	2 h	4 h	12 h
*toll-like receptor 13*	*tlr13*	33.5730	84.0101	252.8270	23.7867
*toll-like receptor 2 type-1*	*tlr2-1*	26.4006	33.7079		
*toll-like receptor 5*	*tlr5*		8963.1958	13,251.6757	27.6041
*myeloid differentiation primary response 88*	*myd88*	62.1266	170.5665	380.9293	
*interleukin-1 receptor-associated kinase 4*	*irak4*		283.4938	349.8690	
*nuclear factor nf-kappa-b p100 subunit*	*nfkb2*		349.8690	142.5494	3.5853
*nf-kappa-b inhibitor alpha*	*nfkbia*	46.3156	554.3636	72.7335	20.6977
*nf-kappa-b inhibitor epsilon*	*nfkbie*	38.6016	54.1131	72.7335	
*map kinase-activated kinase 2*	*mapkapk2*	78.9174	312.7962	458.7292	
*map kinase 3*	*mapk3*		268.6767	444.6294	
*map kinase kinase kinase 4*	*map2k4*	113.5394	171.4468		
*map kinase 12*	*mapk12*	100.5704	169.6868	176.4579	
*map kinase 14*	*mapk14*	69.3530	150.9215	115.8456	10.5737
*map kinase 6*	*mapk6*	36.5802	123.2329	464.0316	
*map kinase 2*	*mapk2*		65.5898		
*map kinase kinase 3*	*map3k3*		32.1681		
*map kinase kinase kinase 4*	*map3k4*		0.0000	5291.6003	
*mitogen-activated kinase kinase kinase 14*	*map3k14*	45.4755	63.9774	190.0614	
*mitogen-activated kinase kinase kinase 8*	*map3k8*	79.1107	137.7203	292.0333	4.8744
*signal transducer and activator of transcription 1*	*stat1*	44.2231	76.8213	145.9930	
*signal transducer and activator of transcription 3*	*stat3*		33.8213	70.3976	
*signal transducer and activator of transcription 4*	*stat4*	16.6058	37.3109	33.8135	
*signal transducer and activator of transcription 6*	*stat6*	46.6821	67.9804	168.4172	
*janus kinase 1*	*jak1*	29.7269	75.7555	238.0845	
*janus kinase 2*	*jak2*	52.6834	33.4907	133.9884	
*suppressor of cytokine signaling 1*	*socs1*	30.3920	54.9888	358.7853	26.6144
*suppressor of cytokine signaling 3*	*socs3*	76.5795	194.8081	787.6507	29.6728

**Table 4 biology-09-00337-t004:** Heat map showing a group of DE genes in turbot peritoneal cells at 1, 2, 4, 12 and 48 hpi, including several genes coding for chemokines, chemokine receptors and enzymes involved in prostaglandin synthesis. Results are expressed as mean expression ratios of groups 1, 2 and 4 hpi vs. 0 h, or 12 and 48 hpi vs. PBS. Red indicates increased gene expression levels; green indicates decreased levels.

Gene Name	Gene Abbrev.	1 h	2 h	4 h	12 h	48 h
*c-x-c motif chemokine ligand 8*	*cxcl8*	696.184767	4267.68045	6208.1874		−4.0735920
*c-c motif chemokine 11*	*ccl11*	126.694565	162.000922	55.8971143		
*c-c motif chemokine 3-like 1*	*ccl3l1*	33.0565132	281.708153	177.056028		
*c-c motif chemokine 4*	*ccl4*	24.6363172	22.0159395		5.04909594	
*c-c motif chemokine 2*	*ccl2*	45.3990169	46.1844289			
*c-c motif chemokine 20*	*ccl20*		448.823876			
*c-x-c chemokine receptor type 4*	*cxcr4*	3607.59607	8791.52888	4637.55324		
*chemokine-like receptor 1*	*cmklr1*	990.041628	1608.67403	3857.3838	6.59418415	
*c-x-c chemokine receptor type 1*	*cxcr1*	835.451583	3172.20243	5777.64259	11.6779436	
*c-x-c chemokine receptor type 2*	*cxcr2*	75.7457614	163.539151	575.328912	27.9524211	
*c-x-c chemokine receptor type 3*	*cxcr3*	249.193869	1113.09829	906.716695		
*c-x-c chemokine receptor type 3-2*	*cxr3-2*	103.928423	81.3886854	26.2114885		
*prostaglandin g h synthase 2*	*ptgs2*	81.9664047	624.749387	418.015344		
*prostaglandin e synthase 3*	*ptges3*	19.3125976				
*prostaglandin i2 synthase*	*ptgis*		80.9131124			
*arachidonate 12s-lipoxygenase*	*alox12*	93.5362347	145.035988	386.109999		
*arachidonate 5-lipoxygenase*	*alox5*				9.32644124	56.2132685
*arachidonate 15-lipoxygenase*	*alox15b*					164.506767
*leukotriene a-4 hydrolase*	*lkha4*	19.7350686	59.5675737	155.71424		
*leukotriene b4 receptor 1*	*ltb4r*	67.2372589	510.252678	866.318345	5.73528823	
*prostaglandin i2 receptor*	*pi2r*	47.2490711				

**Table 5 biology-09-00337-t005:** Heat map showing a group of DE genes in turbot peritoneal cells at 1, 2, 4, 12 and 48 hpi, including genes coding for interleukins, interleukin receptors, genes of the tumour necrosis factor (TNF) family and genes involved in different signalling pathways. Results are expressed as mean expression ratios of groups 1, 2 and 4 hpi vs. 0 h, or 12 and 48 hpi vs. PBS. Red indicates increased gene expression levels; green indicates decreased levels.

Gene Name	Gene Abbrev.	1 h	2 h	4 h	12 h	48 h
*interleukin-1 beta*	*il1b*	327.182241	1589.55462	1847.60352		−6.7387485
*interleukin-16*	*il16*	19.2507463	35.3562406	59.9248328		
*interleukin-27 subunit beta*	*il27b*	13.7300327	242.32254	179.231112		
*myeloid-derived growth factor*	*mydgf*		95.9450925	122.26906		
*interleukin-10*	*il10*				62.3687203	
*interleukin-12 subunit beta*	*Il12b*				11.4185297	23.9473534
*interleukin-6 receptor subunit alpha*	*il6ra*	262.635667	669.959144	1017.60006		
*interleukin-6 receptor subunit beta*	*il6rb*	85.5323549	50.6037764	193.061244		
*interleukin-22 receptor subunit alpha-2*	*il22r2*	122.220327	297.897713	1069.37689		
*interleukin-1 receptor type 1*	*il1r1*	106.471491	238.123796	283.029024	4.98247239	
*interleukin-1 receptor type 2*	*il1r2*	83.0593856	1201.91688	2250.71899	25.847296	
*interleukin-10 receptor subunit alpha*	*il10r1*	101.62706	240.910175	664.168172	9.10643463	
*interleukin-3 receptor class 2 subunit beta*	*il3b2*	66.3168627	197.186399	800.67187	6.3092039	
*interleukin-7 receptor subunit alpha*	*il7ra*	31.0267486				
*interleukin-2 receptor subunit beta*	*il2rb*	28.098746	35.2061352	47.6837501		
*interleukin-2 receptor subunit gamma*	*il2rg*	19.6387686	38.542965	60.4108006		
*interleukin-31 receptor subunit alpha*	*il31r*				3.92933661	
*tumor necrosis factor alpha*	*tnfa*		3614.33852	5953.71784		
*tumor necrosis alpha-induced 8 2 b*	*tnfaip8l2b*	333.798683	408.811225	737.275001		
*tumor necrosis factor ligand superfamily member 6*	*tnfl6*	113.72684	163.681789			
*tumor necrosis factor ligand superfamily member 13b*	*tnfsf13b*	110.928512	52.0480829	271.529023		
*tumor necrosis factor receptor superfamily member 26*	*tnfrsf26*	105.956279	215.164494	196.925793		
*tumor necrosis factor alpha-induced 2*	*tnfaip2*	86.1712612	364.257117	384.174929		
*tumor necrosis factor receptor superfamily member 1b*	*tnfr1b*	65.5284083	57.8161671	64.5415275		
*tnf receptor-associated factor 2*	*traf2*	57.1024509	101.996426	195.25474		
*tumor necrosis factor receptor superfamily member 5*	*tnfrsf5*			41.613748	21.2114576	
*tumor necrosis factor receptor superfamily member 11b*	*tnfrsf11b*				24.2204824	

**Table 6 biology-09-00337-t006:** Heat map showing a group of DE genes in turbot peritoneal cells at 1, 2, 4 and 12 hpi, including several interferon regulatory factor genes. Results are expressed as mean expression ratios of groups 1, 2 and 4 hpi vs. 0 h, or 12 hpi vs. PBS. There were no DE genes at 48 hpi. Red indicates increased gene expression levels; green indicates decreased levels.

Gene Name	Gene Abbrev.	1 h	2 h	4 h	12 h
*interferon-induced 44-like*	*if44l*	71.9994179	111.460711	150.105919	12.8949636
*interferon-induced helicase c domain-containing 1*	*ifih1*	27.3095849	27.3256044	32.2526372	
*interferon- double-stranded rna-activated kinase*	*eif2ak2*	49.9656668	106.466172	196.446601	196.446601
*interferon-related developmental regulator 1*	*ifrd1*	22.9845565	31.2624166	75.4855289	
*interferon-induced 35 kda homolog*	*ifi35*	18.9428499	34.0369582	47.5850614	
*interferon alpha beta receptor 2*	*ifnar2*	74.1505791	71.2747429	76.2223874	
*interferon regulatory factor 1*	*irf1*	14.0519125	18.1839762		10.9989039
*interferon regulatory factor 2*	*irf2*	24.2396496	43.3669569	81.4920354	
*interferon regulatory factor 3*	*irf3*	22.1667324	54.0067235	79.6835222	
*interferon regulatory factor 8*	*irf8*	10.1543906	24.2065428	63.6000421	4.01348333
*interferon-induced gtp-binding mx*	*mx*	15.4202932	37.2305381	68.2243287	
*interferon-induced with tetratricopeptide repeats 1*	*ifit1*		18.5057029	57.1941474	
*interferon regulatory factor 5*	*irf5*		21.2670525	31.5542826	
*interferon-induced very large gtpase 1*	*gvinp1*			95.658683	5.53328268

**Table 7 biology-09-00337-t007:** Heat map showing a group of DE genes in turbot peritoneal cells at 1, 2, 4 and 12 hpi, including many genes involved in cell death. Results are expressed as mean expression ratios of groups 1, 2 and 4 hpi vs. 0 h, or 12 hpi vs. PBS. There were no DE genes at 48 hpi. Red indicates increased gene expression levels; green indicates decreased levels.

Gene Name	Gene Abbrev.	1 h	2 h	4 h	12 h
*apoptosis facilitator bcl-2 14*	*bcl2l14*	113.684	898.176	1577.930	
*bcl2 associated agonist of cell death*	*bad*	189.050	352.048	452.689	
*apoptosis regulator bax*	*bax*	32.616	72.452	176.064	4.928
*caspase recruitment domain-containing 11*	*card11*	16.892	32.920	40.394	
*cysteine serine-rich nuclear 1*	*csrn1*	223.940	974.630	2534.320	
*DNA fragmentation factor subunit beta*	*dffb*	45.668	44.024	110.253	
*programmed cell death 10*	*pdcd10*	25.266	167.783	67.591	
*programmed cell death 6*	*pdcd6*	41.736	95.197	162.973	
*programmed cell death 6-interacting*	*pdcd6i*	30.355	84.740	67.733	
*programmed cell death 4*	*pdcd4*	68.163	110.907	162.973	−3.435
*programmed cell death 1 ligand 1*	*pd1-l1*			503.383	64.837
*serine threonine- kinase 17a*	*st17a*	31.071	93.784	201.549	
*serine threonine- kinase 17b*	*st17b*	56.748	56.748	251.152	
*fas cell surface death receptor*	*faslgr*	129.441	119.114	339.304	
*fas-associated death domain*	*fadd*	174.716	313.210	515.828	132.765
*apoptosis-associated speck containing a card*	*pycard*		16.474		
*baculoviral iap repeat-containing 2*	*birc2*		122.763	233.354	3.976
*caspase recruitment domain-containing 9*	*card9*			6092.587	
*caspase-3*	*casp3*		158.260	327.980	
*casp8 and fadd-like apoptosis regulator*	*cflar*				6.554
*caspase-8*	*casp8*				5.508
*fas apoptotic inhibitory molecule 3*	*faim3*				132.765
*b-cell lymphoma leukemia 10*	*bcl10*	67.237	278.829	504.966	
*pyrin*	*mefv*		102.595	208.050	
*receptor-interacting serine threonine- kinase 2*	*ripk2*	16.727	95.564	103.176	
*receptor-interacting serine threonine- kinase 3*	*ripk3*	31.089	64.526	110.020	

**Table 8 biology-09-00337-t008:** Heat map showing a group of DE genes in turbot peritoneal cells at 1, 2, 4 and 12 hpi, including several genes related to B cells and cytotoxic T cells. Results are expressed as mean expression ratios of groups 1, 2 and 4 hpi vs. 0 h, or 12 hpi vs. PBS. There were no DE genes at 48 hpi. Red indicates increased gene expression levels; green indicates decreased levels.

Gene Name	Gene Abbrev.	1 h	2 h	4 h	12 h
*high affinity immunoglobulin epsilon receptor subunit gamma*	*fcerg*	176.817257	311.445965	598.32369	
*immunoglobulin heavy constant mu*	*ighm*	44.659176	56.1328349	52.6153499	
*immunoglobulin lambda constant 6*	*iglc6*	109.625362	108.139186	95.911698	
*immunoglobulin lambda variable 7-46*	*iglv7-46*	49.9266366	61.4466313		
*v-type immunoglobulin domain-containing suppressor of t-cell activation*	*vsir*	169.315053	314.679434	756.121188	
*granzyme a*	*graa*	49.9266366	116.08526	60.9648276	
*granzyme b*	*grab*	169.315053			
*perforin-1*	*perf*	8033.37216			
*plastin-2*	*lcp1*	26.9806006	60.8179271	112.085751	
*immunoglobulin delta heavy chain*	*igd*	106.748406	166.358035		
*b lymphocyte-induced maturation 1*	*blimp*		375.643759	652.345739	
*lymphocyte cytosolic 2*	*lcp2*		42.2394554		
*immunoglobulin superfamily member 3*	*igsf3*			4813.39137	
*b-cell receptor cd22*	*cd22*				5.21002715
*cd83 antigen*	*cd83*	33.9279207	157.493124	189.841453	
*b-cell antigen receptor complex-associated alpha chain*	*cd79a*		38.4830819	58.4676507	

**Table 9 biology-09-00337-t009:** Heat map showing a group of DE genes in turbot peritoneal cells at 1, 2, 4 and 12 hpi, including complement and coagulation related genes. Results are expressed as mean expression ratios of groups 1, 2 and 4 hpi vs. 0 h, or 12 hpi vs. PBS. There were no DE genes at 48 hpi. Red indicates increased gene expression levels; green indicates decreased levels.

Gene Name	Gene Abbrev.	1 h	2 h	4 h	12 h
*coagulation factor viii*	*f8*	24.9906392	126.629374	84.4903238	33.3428858
*coagulation factor xiii a chain*	*f13a1*	144.605804	117.464967	111.905425	
*tissue factor pathway inhibitor*	*tfpi1*	32.3367549	59.5824002	52.4958644	
*tissue factor*	*tf*				106.435028
*complement c1q subcomponent subunit b*	*c1qb*	27.8372678	54.1585349	31.7288673	
*complement c1s subcomponent*	*c1s*	76.2220545	76.8459537		
*complement factor d*	*cafd*	115.641444	434.049445		
*complement factor h*	*cafh*	68.4428124	112.403061	39.3661006	
*complement c3*	*c3*			51.9446282	
*complement c4*	*c4*	28.5701165	52.4312679	73.1753333

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
