# Peer review of "Interactions between the Parasite *Philasterides dicentrarchi* and the Immune System of the Turbot *Scophthalmus maximus*. A Transcriptomic Analysis"

_biology, 2020, doi:10.3390/biology9100337_

Round 1
Reviewer 1 Report
In the present manuscript, Alejandra Vale with colleagues aimed to investigate the molecular mechanisms underlying the interaction between the immune system of turbot and its parasite Philasterides diecentrachi using the RNAseq. This holistic approach to host-pathogen interaction and the Herculean effort of the team resulted in a well-rounded manuscript, providing valuable insights to the field of fish parasitology and immunology.
The introduction provides sufficient background to the readers, the methods are described well and the results are well structured and presented. The discussion exceeds four pages, but it is very informative and discusses most of the recent literature.
I endorse the publication of the manuscript after resolving some minor typos and editing errors. Also, it would be helpful if authors dedicate a couple of sentences in the introduction to the use of the peritoneal model of the injection, as this doesn't seem to be well explained. Other issues include :
- In line 127 authors talk of antigen preparation which is slightly confusing. I would suggest using "parasite" instead.
- line 222 - glyceraldehyde-3-phosphate dehydrogenase is in different size of the font
- line 225,226,227 references are underlined, as is the P. dicentrarchi in line 301
- in Figure 2 - there is a "Plot area" sign in the figure
- line 479, 480 the references are in different size font
- line 587 "...and at 1, 3, 4 and 12 hpi....there is no 3hpi timepoint in this study
- throughout the manuscript, I would recommend substituting the "early moments of infection" with "early stages of infection"
Author Response
Thank you very much for reviewing the manuscript and for your comments. We agree with all the suggestions proposed. The number of the lines correspond to the reviewed manuscript.
In the present manuscript, Alejandra Vale with colleagues aimed to investigate the molecular mechanisms underlying the interaction between the immune system of turbot and its parasite Philasterides dicentrachi using the RNAseq. This holistic approach to host-pathogen interaction and the Herculean effort of the team resulted in a well-rounded manuscript, providing valuable insights to the field of fish parasitology and immunology.
The introduction provides sufficient background to the readers, the methods are described well and the results are well structured and presented. The discussion exceeds four pages, but it is very informative and discusses most of the recent literature.
- I endorse the publication of the manuscript after resolving some minor typos and editing errors. Also, it would be helpful if authors dedicate a couple of sentences in the introduction to the use of the peritoneal model of the injection, as this doesn't seem to be well explained.
We have added the following sentence at the end of the introduction (lines 122-125):
Parasites were injected intraperitoneally and the transcriptomic response was evaluated in both fish cells and parasites found in the peritoneal cavity at several different times. This infection model enabled us to obtain sufficient numbers of host cells and parasites for the analysis.
Other issues include :
- In line 127 authors talk of antigen preparation which is slightly confusing. I would suggest using "parasite" instead.
Line 142. We agree with the suggestion.
- line 222 - glyceraldehyde-3-phosphate dehydrogenase is in different size of the font
Line 240. Corrected
- line 225,226,227 references are underlined, as is the P. dicentrarchi in line 301
Lines 243, 244, 250, 319. Corrected
- in Figure 2 - there is a "Plot area" sign in the figure
It does not appear in the original figure
- line 479, 480 the references are in different size font
Lines 497, 499. Corrected
- line 587 "...and at 1, 3, 4 and 12 hpi....there is no 3hpi timepoint in this study
Line 607. Corrected. It is 1, 2 and 4.
- throughout the manuscript, I would recommend substituting the "early moments of infection" with "early stages of infection"
We agree that it is more appropriate using stages. We have revised all the document and we have replaced early moments with early stages.
Reviewer 2 Report
This manuscript shows information about early response of peritoneal cells of turbot to an experimental infection with P. dicentrarchi, as well as the transcriptomic response in the parasite. Gathering all this kind of data is very important to understand the host-parasite interactions and will contribute to ameliorate the knowledge of this relevant disease. However, there are some aspects that can be improved. In order of appearance:
Line 115: L as symbol of litre; check the symbol of Celsius degrees (also line 152).
Lines 129-130: Check letter size and font. "P. dicentrarchi" should be either not in italics or underlined.
Lines 131-144: This paragraph may be summarised. There is too many sentences to explain that parasites survive in the cavity when injected at 107 dose compared to a dose of 105, in which they die. Which is the purpose of the lower dose in the second experiment? Maybe the authors could find more differentially-expressed genes in turbot at 12 and 48 hpi if the fish were injected with the higher dose.
Line 135: "engulf" instead of "phagocytose".
Line 192 and throughout the text: Define abbreviations the first time mentioned in the manuscript. I.e. DE, UR, DR...
Lines 207-208: These information should be shown in "Experimental infection" subsection.
Line 219: Check double spaces. Also in line 222, 548, 551.
Line 222: Check letter size and font. Also in line 225, 479, 480.
Lines 233-236: Information provided in subsection 2.9 does not fit with the manuscript. Remove this section.
Line 251: Check thousand separator. The point is a decimal separator. It is preferable to use a space: "99 821" instead of "99.821". Also in line 258.
Lines 266-269: Subsections 3.1 and 3.2 may be fused into: RNAseq analysis and validation.
Line 301: "P. dicentrarchi" and the space between "ABC transporters" should not be underlined.
Line 305: This sentence fits better in the Discussion section. Also line 309, 315, 353-354, 369, 416, 444.
Line 324: The space between "P. dicentrarchi" should not be underlined.
Line 328: "P. dicentrarchi" should be underlined.
Lines 397-402: Revise whether "tnf" should be capitalized or in italics when referred to genes.
Line 432: IgM and IgD immunoglobulins
Lines 417-418: "Several" twice. Also in 445-446.
Line 484: The sense of the sentence should be the other way around, the results of this study are in agreement with previous works.
Line 558: "TLRs" instead of "TLRS".
Line 563: "and" should not be in italics.
Line 587: "IL10" instead of "il10".
Line 591: "IL12B" instead of "il12b".
Line 605: "irf8" instead of "IRF8".
Line 630: 1 hpi
Line 657: Transcriptomic analysis
Lines 661-663: Consider using abbreviations for days post-infection.
Line 695: Lacks a space before "This" and letter T should not be in bold.
Figures: The font in the figures is different from the manuscript. Figure 2 seems to be a screenshot, as it has a text box "Plot Area" over the graphic. Check the format of the legend of Figure 1. Line 282: avoid abbreviation DE. Line 283: P. dicentrarchi in italics.
Tables: The decimal separator should be a point instead of a comma. As a suggestion, to improve the format, suprime "48 h" column in tables 3, 6, 7, 8, and 9. Authors may indicate in the legends that there were not DE genes at these time post exposure. If not, check the heathers and the legends of the tables and supplementary material, sometimes it is mentioned "48 h" and other "24 h", i.e. tables 6 and 8, line 680.
Author Response
Thank you very much for the meticulous reviewing the manuscript and for the comments. We agree we agree with all the suggestions and comments. The number of the lines correspond to the reviewed manuscript.
This manuscript shows information about early response of peritoneal cells of turbot to an experimental infection with P. dicentrarchi, as well as the transcriptomic response in the parasite. Gathering all this kind of data is very important to understand the host-parasite interactions and will contribute to ameliorate the knowledge of this relevant disease. However, there are some aspects that can be improved. In order of appearance:
- Line 115: L as symbol of litre; check the symbol of Celsius degrees (also line 152).
Lines 129 and 168. Corrected
- Lines 129-130: Check letter size and font. "P. dicentrarchi" should be either not in italics or underlined.
Line 145. Corrected
- Lines 131-144: This paragraph may be summarised. There is too many sentences to explain that parasites survive in the cavity when injected at 107 dose compared to a dose of 105, in which they die. Which is the purpose of the lower dose in the second experiment? Maybe the authors could find more differentially-expressed genes in turbot at 12 and 48 hpi if the fish were injected with the higher dose.
Lines 145-151 and 159-160. We have shortened the paragraph, as suggested by the reviewer. The lower ciliate concentration was used because most fish cells would be engulfed, and probably killed by the parasite at 12 and 48 hpi if fish were injected with the higher concentration. We have added this information to the paragraph “This ciliate concentration was selected because most fish cells would be engulfed and killed at 12 and 48 hpi if the higher concentrations were used”.
- Line 135: "engulf" instead of "phagocytose".
Line 160. We have engulf.
- Line 192 and throughout the text: Define abbreviations the first time mentioned in the manuscript. I.e. DE, UR, DR...
We had defined those abbreviations in the abstract. Now, also in lines 271, 292, and 308
- Lines 207-208: This information should be shown in "Experimental infection" subsection.
Lines 175-176. We have moved this information to “Experimental infection” subsection.
- Line 219: Check double spaces. Also in line 222, 548, 551.
Checked
- Line 222: Check letter size and font. Also in line 225, 479, 480.
Lines 243, 245, 478 and 479. Checked
- Lines 233-236: Information provided in subsection 2.9 does not fit with the manuscript. Remove this section.
Lines 251-254. We have maintained the statistical analysis because it was used to determine differences between groups in the number of ciliates or in the number of fish cells in the peritoneal cavity (supplementary files).
- Line 251: Check thousand separator. The point is a decimal separator. It is preferable to use a space: "99 821" instead of "99.821". Also in line 258.
Lines 269 and 276. As suggested, we have separated thousands in large numbers with a space.
- Lines 266-269: Subsections 3.1 and 3.2 may be fused into: RNAseq analysis and validation.
Line 256. As suggested, we have fused the two subsections.
- Line 301: "P. dicentrarchi" and the space between "ABC transporters" should not be underlined.
Line 319. Corrected
- Line 305: This sentence fits better in the Discussion section. Also line 309, 315, 353-354, 369, 416, 444.
Lines 327, 333, 334, 371-373, 395-396, 435-436, 463-465. We have revised those sentences and they were were reduced or eliminated, as they were also included in the Discussion (not the same sentence by the same idea)
- Line 324: The space between "P. dicentrarchi" should not be underlined.
Line 342. Revised
- Line 328: "P. dicentrarchi" should be underlined.
Line 346. Revised
- Lines 397-402: Revise whether "tnf" should be capitalized or in italics when referred to genes.
Line 423. It should be in cursive. Revised
- Line 432: IgM and IgD immunoglobulins
Line 451. We have deleted immunoglobulins
- Lines 417-418: "Several" twice. Also in 445-446.
Lines 437 and 465. We have used a different word
- Line 484: The sense of the sentence should be the other way around, the results of this study are in agreement with previous works.
Line2 505-504. We agree with the suggestion and we have modified the sentence
- Line 558: "TLRs" instead of "TLRS".
Line 578. Revised
- Line 563: "and" should not be in italics.
Line 583. Revised
- Line 587: "IL10" instead of "il10".
Line 607. Revised
- Line 591: "IL12B" instead of "il12b".
Line 611. Revised
- Line 605: "irf8" instead of "IRF8".
Line 625. Revised
- Line 630: 1 hpi
Line 650. Revised
- Line 657: Transcriptomic analysis
Line 677. Revised
- Lines 661-663: Consider using abbreviations for days post-infection.
Line 683. We have used dpi
- Line 695: Lacks a space before "This" and letter T should not be in bold.
Line 716. Revised
- Figures: The font in the figures is different from the manuscript. Figure 2 seems to be a screenshot, as it has a text box "Plot Area" over the graphic. Check the format of the legend of Figure 1. Line 282: avoid abbreviation DE. Line 283: P. dicentrarchi in italics.
Plot area does not appear in the original figure. Lines 301-301. The format of the legend has been checked and P. dicentrarchi in italics.
- Tables: The decimal separator should be a point instead of a comma. As a suggestion, to improve the format, suprime "48 h" column in tables 3, 6, 7, 8, and 9. Authors may indicate in the legends that there were not DE genes at these time post exposure. If not, check the heathers and the legends of the tables and supplementary material, sometimes it is mentioned "48 h" and other "24 h", i.e. tables 6 and 8, line 680.
We have taken all of these suggestions into account. We have revised the tables and deleted the 48 h column in those cases in which there were no DE genes at that time point (tables 3, 6, 7, 8 and 9), and we have added to the legend that “There were no DE genes at 48 hpi). We have revised all the tables and used a point instead of a comma as a decimal separator. We have also revised the tables of supplementary material (line 701)
Reviewer 3 Report
This study analyzed the transcriptomic profiles of the parasites and the hosts during infection of the P. dicentrarchi on turbot. A great deal of effort has been paid regarding an extensive list of genes differentially expressed in the parasite as well as in the host fish during the early stages of parasite infection. This is interesting topic and the results will be helpful for understanding the mechanism behind interaction between the parasite and fish. I recommend the MS can be accepted after revision. Detailed suggestions are as follows:
General comments:
- The authors argue that they identified genes involved in virulence or in resistance to the host immune system. This may be the case if there are only parasites that can survive from the infection process. However, can we make the same conclusion if there are transcripts from parasites dying in the process? In this regard, are we sure that number shown in Fig.S2B reflect the ciliates that are alive?
- Regarding values (mean expression ratios) in Tables 1-9, are these values from how many samples? Did all the fish/parasites in the group showed the same DE pattern?
- Regarding some genes (e.g. cxcl8 in Table 4 , il1b in Table 5, and pdcd4 in Table 7) showing an opposite pattern of expression, - can we rely on the data showing an opposite pattern in one sample?- For examples, L371-2 Cxcl8 was strongly expressed at 1, 2 and 4 hpi, but was DR at 48 hpiL569 Among these, the strongest regulation was found in cxcl8…L391/2 The pro-inflammatory il1β was strongly UR at 1, 2 and 4 hpi and, similarly to cxcl8, was DR at 48 hpi.
- How should we interpret the results showing that almost all of differentially expressed genes in peritoneals cells (Tables 3-9) are up regulated?
- Other minor comments- inconsistencies in designating P. dicentrarchi in Figure 1, Tables 1 & 2 legends
- L211-212 Nano Drop, Nano..
- L234 : The values shown in the figures are means ± SD. Is this right?
- L337 immune response (4.3 to 4.8%)… check the number
- L 220 & L690, supplementary Fig S4 ccl11b?
- L558 TLRs
Author Response
Thank you very much for reviewing the manuscript and for your comments. We have considered all of the suggestions and made the relevant changes or comments. The number of the lines correspond to the reviewed manuscript.
This study analyzed the transcriptomic profiles of the parasites and the hosts during infection of the P. dicentrarchi on turbot. A great deal of effort has been paid regarding an extensive list of genes differentially expressed in the parasite as well as in the host fish during the early stages of parasite infection. This is interesting topic and the results will be helpful for understanding the mechanism behind interaction between the parasite and fish. I recommend the MS can be accepted after revision. Detailed suggestions are as follows:
General comments:
- The authors argue that they identified genes involved in virulence or in resistance to the host immune system. This may be the case if there are only parasites that can survive from the infection process. However, can we make the same conclusion if there are transcripts from parasites dying in the process? In this regard, are we sure that number shown in Fig.S2B reflect the ciliates that are alive?
R. Living ciliates can be easily identified and we believe that Fig. S2B accurately reflects the number of living ciliates in peritoneal samples. Dead ciliates stop moving and lose their shape, becoming rounded, and are lysed very quickly (in minutes) (all these changes can be easily observed in the microscope). For these reasons, we believe that most cytoplasmic transcripts of those ciliates would be eliminated during sample preparation. We cannot rule out the possibility that some ciliates in the samples are dying, although, on the basis of the results shown in Fig. S2B, we expect the percentage to be very low.
2. Regarding values (mean expression ratios) in Tables 1-9, are these values from how many samples? Did all the fish/parasites in the group showed the same DE pattern?
R. As indicated in M&Ms, lines 147 and lines 157 (now lines 150 and 160, respectively), 3 replicates of three fish each per time point were used in the first experiment (0, 1, 2 and 4 hpi), and three replicates of 7 fish each were analysed at each time point in the second experiment (12 and 48 hpi). The response was very similar across replicates, especially for the genes that are highly regulated; i.e. ABC transporter G family and genes of the leishmanolysin family, in the case of Philasterides, or genes related to the inflammatory response, in the case of fish cells.
3. Regarding some genes (e.g. cxcl8 in Table 4 , il1b in Table 5, and pdcd4 in Table 7) showing an opposite pattern of expression, - can we rely on the data showing an opposite pattern in one sample?- For examples, L371-2 Cxcl8 was strongly expressed at 1, 2 and 4 hpi, but was DR at 48 hpiL569 Among these, the strongest regulation was found in cxcl8…L391/2 The pro-inflammatory il1β was strongly UR at 1, 2 and 4 hpi and, similarly to cxcl8, was DR at 48 hpi.
R. The response obtained in genes involved in the inflammatory response, such as the chemokine cxcl8 or the proinflammatory cytokine Il1b, was very consistent. Expression of these genes increased very quickly during the early stages of infection and then decreased when infection was controlled. In the case of pdcd4, which is involved in apoptosis, the results obtained also suggest this process occurs very early on, at least in some of the cells. These processes are not mutually exclusive. In previous studies, we found that the parasite induces strong migration of cells to the peritoneal cavity and that many of those cells are highly stimulated by the parasite - probably as a result of the interaction of parasite cilia (which seems to have proteases at the cell membrane) with leucocytes and also because many fish cells are engulfed by the parasite. However, many of those engulfed fish cells probably activate apoptosis as a consequence of the release of lysosomes to the digestive vacuole by the ciliate.
4. How should we interpret the results showing that almost all of differentially expressed genes in peritoneals cells (Tables 3-9) are up regulated?
R This was probably because the number of leukocytes in the peritoneal cavity of fish injected with parasites increased greatly during infection. However, the number of living parasites probably remained stable.
5. Other minor comments- inconsistencies in designating P. dicentrarchi in Figure 1, Tables 1 & 2 legends
Lines 300, 319, 344. Revised
- L211-212 Nano Drop, Nano..
Lines 229-230. Revised
- L234 : The values shown in the figures are means ± SD. Is this right?
It is correct
- L337 immune response (4.3 to 4.8%)… check the number
Line 355. Checked
- L 220 & L690, supplementary Fig S4 ccl11b?
cd11b is correct
- L558 TLRs
Line 578. Checked